# Qualitative analysis from the social referents perspective of the multidimensional construct of schoolchildren's motor competence

Manuel Segura-Berges[1], Carlos Peñarrubia-Lozano[2], Manuel Lizalde-Gil[2†], Juan Carlos Bustamante[3]*

1 Colegio Compañía de María, Zaragoza, Spain, 2 Department of Musical, Plastic and Corporal Expresion, University of Zaragoza, Zaragoza, Spain, 3 Department of Psychology and Sociology, University of Zaragoza, Zaragoza, Spain

☉ These authors contributed equally to this work.
† Deceased.
* jbustama@unizar.es

**Data Availability Statement:** All relevant data are within the paper and its Supporting Information files. However, the dataset and transcripts that

## Abstract

Motor competence (MC) as a multidimensional construct is influenced by motor, cognitive, emotional and social variables. It is also determined by schoolchildren interacting with their own context. Thus pre-adolescence is a sensitive stage in development when physical, emotional and cognitive changes are manifested. By taking this context in accountn, the perception of the social referents close to schoolchildren allows for a deeper understanding of the role and influence of all these variables in a broader MC concept. For this purpose, a qualitative study was conducted by discussion groups and semistructured interviews, respectively, for teachers and family members in Primary Education. The content analysis was carried out according to the main study dimensions, associated with the motor, cognitive, affective-emotional and social domains. Both family members and teachers point out that the affective-emotional level has a stronger impact on the development of motor skills than the motor level. For children to develop their MC, it is necessary to create appropriate contexts in which family members and teachers are the main agents of influence. In conclusion, assessing MC in the educational sphere must have a more comprehensive and broad approach. It is necessary to bear in mind a larger number of variables involved in schoolchildren's motor development to make the most objective assessment possible and, likewise, to promote facilitating environments that help their development.

## Introduction

The aim to study motor competence (MC) in schoolchildren lies in investigating the implication of motor, emotional, cognitive and social variables in the multidimensional concept of this construct [1, 2]. In line with this, reference state law on education for the context in which this work was conducted [3] also points out the importance of MC in the education context in both motor conduct development terms and cognitive, social and emotional domains. As for

underlie the results described in our manuscript are available from Figshare data repository (https://figshare.com/articles/dataset/Data_Qualitative_analysis_from_the_social_referents_perspective_of_the_multidimensional_construct_of_schoolchildren_s_motor_competence_/20791534)

**Funding:** This research has been supported by grants received by CPL from the Fundación Bancaria Ibercaja (JIUZ-2020-SOC-15). The funders had no role in study design, data collection and analysis, decision to publish, or preparation of the manuscript.

**Competing interests:** The authors have declared that no competing interests exist.

the motor dimension, coordination is established as one of the backbones of MC, which is, in turn, outlined as a complex function that manifests itself by acting in the different facets of our lives [4]. In parallel, the cognitive plane is also related to the motor actions that schoolchildren present. Hence perceived MC is shaped as individuals' awareness and belief in their ability to perform fine and gross motor tasks. This perceived competence generates a series of feelings that are experienced when the results are positive and the perception of competence in a task is a stimulus to gain confidence and to repeat it [5]. Individuals' beliefs in their operational capabilities function as a set of close determinants of their behavior, their thought patterns and the emotional reactions that they experience in difficult situations. Fort-Vanmeerhaeghe et al. [6] defend the need to take into account the psycho-emotional dimension through variables such as anxiety and self-esteem, which are two variables that are, in principle, antagonistic and allow to know the subject' state when facing a motor situation. Finally, all these variables are also conditioned by social variables. In addition, Vedul-Kjelsås et al. [7] showed that the relations of these dimensions vary during the development of schoolchildren's MC vary according to gender.

Previous consulted studies about MC pay attention especially to the motor component [8, 9]. Thus some studies deem it necessary to address MC from the multidimensional point of view by not only covering motor factors, but also cognitive, psychosocial and social factors [10, 11]. This more integrally promotes a teaching-learning process in the motor domain, and in accordance with the development characteristics that schoolchildren present from infancy to adolescence [12, 13]. Therefore, analyzing MC in schoolchildren must not be merely applied to the motor domain, but should also cover more variables given its multidimensional nature [14, 15]. In this way, and based on this holistic vision of the MC concept, schoolchildren's contextual factors and social referents seem to play a key role and strongly influence their understanding of MC development [4, 16–18].

According to previous studies, the stage between 9 and 12 years old is when the greatest motor development occurs from the evolutionary and structural points of view [12, 13, 19]. It is also a time when cognitive and emotional aspects are especially important in personality development when physical and psychological changes occur which, in turn, transform their biology, behavior and social relationships [8, 20–22]. Along these lines, previous studies refer to the importance of the perception of schoolchildren's significant and close people to deal with these children's MC concept at pre-adolescent ages, whose family members and teachers are the main referents [23–26].

Nonetheless, Laukkanen et al. [27] evidence that family members and teachers' perception of schoolchildren's MC development has been mostly and almost exclusively studied in the motor domain, which makes the interpretation and perception of a more global MC construct very difficult. By bearing in mind these considerations, and in agreement with Kennedy et al. [28], creating a broader and more global and complete image of the factors that impact children's real MC by contemplating the perceptions of both schoolchildren and their close social referents seems to make sense.

In light of all this, the present study attempts to investigate through a qualitative approach the perception of schoolchildren's close social referents, like family members and teachers, of the role and relevance of the motor, social, emotional and cognitive variables in a broader school MC concept in the pre-adolescent stage. With this objective, the novelty of the study lies in investigating the set of dimensions involved in the more global MC concept, and not only in the motor dimension. In this way, it allows to deepen in the experience and perspective of the significant close models, relatives and teachers that can influence the development of MC at primary ages [27].

## Materials and methods

### Participants

A qualitative methodology was followed by taking the closest social referents of Primary Education pupils aged between 9 and 12 years as the study reference framework. Thus the study involved teachers from different subject areas between grades 4 and 6 grade of Primary Education and schoolchildren's family members at these same education levels (Table 1), which correspond to these age groups. A convenience sampling process [29, 30] was used to set up the sample, which took into account the following variables: gender and parental role (woman-mother/man-father); the grade in which teachers work or, in the case of family member, that in which schoolchildren study (grades 4, 5 and 6); type of education institution (Public/Private). Participants were informed of the study objectives and voluntarily agreed to participate after having verified that they met the inclusion criteria.

There were 19 participants in the Teacher category (average age 40.47±10.80 years): 11 men (58.89%) and 8 women (41.11%); 8 (41.11%) belonged to Public and 11 (58.89) to Private schools. Finally, 7 (36.84%) taught grade 4, 6 (31.57%) grade 5 and the last 6 (31.57%) grade 6. There were 38 participants in the family members category (42.50±2.83 years), 21 women (55.26%) and 17 men (44.73%). Concerning the ownership of the education centre, participants were equally divided between Public and Private centers (n = 19, 50% in each group). According to courses, 14 family members (36.84%) belonged to grade 4, 13 (26.31%) to grade 5 and 11 (28.94%) to grade 6. No statistically significant differences (p>0.05) were found in the distribution of the variables used for sampling.

### Ethics

This study was appraised favorably by the Clinical Research Ethics Committee of Aragón (Spain) in its Minutes N.: 14/2020. Participation in the study was voluntary and based on written informed consent for both family members and teachers.

### Data collection procedure

The presented qualitative data were collected between November 2020 and January 2021, which coincided with the end of the first trimester and the start of the second trimester of academic year 2020/2021. The data collection process was carried out in the same way and by the

**Table 1. Participants.**

| Social referents | | Teachers | | Families | |
|---|---|---|---|---|---|
| Data Collection Technique | | Discussion group | | Interviews | |
| Gender | | M[a] | W[b] | M[a] | W[b] |
| Public School | 4th PEd[c] | 2 | 2 | 2 | 4 |
| | 5th PEd[c] | 2 | 1 | 4 | 2 |
| | 6th PEd[c] | 3 | 1 | 4 | 3 |
| Private School | 4th PEd[c] | 1 | 2 | 2 | 6 |
| | 5th PEd[c] | 2 | 1 | 3 | 4 |
| | 6th PEd[c] | 1 | 1 | 2 | 2 |
| | Total | 11 | 8 | 17 | 21 |

[a]Men.

[b]Women.

[c]Primary Education.

same researcher. An email was first sent with an invitation to participate in the study. Next a Google Forms link was forwarded with informed consent and to request data to identify the sample.

A qualitative and interpretative approach, and discussion group techniques and individual semistructured interviews [31], were chosen to meet the study objective. Regarding structure, both techniques included a series of questions about the motor, social, cognitive and psychosocial domains that shape a more global MC concept [32, 33]. Having reviewed the previous literature, a three-level system of categories (dimensions, categories, indicators) was generated (S1 File) based on the four variables mentioned above. This classification tree, together with its definitions, was reviewed by an external person to the study, an expert in qualitative methodology and with a professional background related to the Physical Education (PE) area. Having configured the definitive system of categories, an initial script of questions was designed to conduct interviews with family members and discussion groups with teachers. Once again, the work of the external consultant was required, who made appreciations such as changes in the order of questions or nuances in the wording of some of them. Finally, 10 open questions were designed for both the discussion groups and semistructured interviews, and were employed for both teachers and family members. This consisted in answering questions like: "*With what type of emotional barriers or difficulties does a child come across when performing motor actions in his/her context*?" or "*What motor capacities do you think are more related to children moving well while performing physical activities? Why?*" (S2 File).

With family members, individual semistructured interviews were held to ensure that encounters were flexible and dynamic, during which emerging themes were expected that would have to be explored. Interviews were conducted using Microsoft Skype and Google Meet, and their average duration was 25 min. For teachers, the discussion group technique was chosen to collect data because it allows their perceptions based mainly on their training, their teaching style and their social action context to be ascertained and combined [34–36]. Discussion groups were conducted using Google Meet. Their average duration was 40 min. Focus groups were formed randomly by taking into account educational centre ownership criteria (public/private) and the grade in which the teachers worked. In all cases (interviews and focus groups), the study objectives were recalled by verbally confirming the consent of participants to be involved in it.

This data collection strategy for both groups allowed the views of the different groups participating in the study to be compared or triangulated. The Consolidated Criteria for Reporting Qualitative Research (COREQ) checklist with further details of the qualitative design is included in the (S3 File).

## Analyses

A content analysis of the reported responses was deductively carried out. The initial category tree was constructed following a deductive logic [37, 38] based on previous literature. Version 12 Plus of the Nvivo software was used to do the analysis (https://www.qsrinternational.com/nvivo/home).

A concordance analysis was performed of the two researchers regarding interviews (n = 9, 23.68%) and discussion groups (n = 1, 25.00%). No emerging categories appeared in this first analysis. The obtained Kappa coefficient was k = .90. A second intrapersonal concordance analysis was performed. A coefficient of k = .88 was obtained, which allowed the analysis to be completed by only one researcher.

The study dimensions are defined below (the operational definitions of the information units forming the main classification tree are found in S1 File):

- The main point of interest of the *motor level* lies in the physical capacities (both conditional and perceptive-motor), and the basic and specific motor skills, which social referents believe to be more related to good motor capacity when schoolchildren perform motor actions

- The *cognitive level* refers to the self-perception motor construct that focuses on knowing what social referents believe is the self-perception type of MC that is more linked with motor-competent schoolchildren; it is either perception based on motor actions for which only the body is used or that which requires using mobile objects to perform it

- The *affective-emotional level* centers mostly on knowing social referents' perception of the psychosocial implications in schoolchildren's MC concept by bearing in mind the attitudes and feelings that they display while developing MC in various contexts

- The *social level* focuses mainly on knowing the social support of the people close to schoolchildren to help them to perform physical activity, plus the presence of models and programs that promote performing physical activity to, thus, contemplate contexts in which pupils can interact with the environment and their peers by practicing physical activity

## Results

The quantitative descriptive analysis with the obtained results is shown in Table 2. To guide the interpretative analysis, and in accordance with one of our previous studies [29], the descriptive data analysis includes the number of codifications of each category, their presence in the different analyzed documents and the extent of each one (measured in lines).

The qualitative analysis is presented below. It contains textual extracts taken from different interviews and discussion groups to reinforce the obtained results. To ensure participants' anonymity, they were coded with two elements: a descriptor of education agents (FAM, family members/MAE, teachers) and a number (01 to 38).

### Motor level

In category *1.1 Coordination and balance*, both family members and teachers refer to these capacities being more related to the motor dimension of the MC concept. They stress that this type of capacities allows pupils to fluently perform quality overall motor execution: *I think that good motor performance is related mainly to coordination because I consider coordination to be the cogs that enable other capacities to work properly* (FAM 15). Teachers point out the importance of bearing in mind the perceptive nature of this capacity when contemplating internal and external stimuli:

> That's not all because, on the other hand, to well coordinate movements, or to combine movements with others with plasticity, it is necessary to consider the exterior stimuli that do not form part of our body so our body can do what we really want it to do (MAE 04).

Both agents also emphasize the importance of perceptive motor capacities, such as coordination and balance, for the complex psycho-sensorial adjustments required when applying neuromuscular skills: *I think that balance is basic to move well both statically and while moving because balance allows the body to be erected and to adopt an optimum position to perform movement* (MAE 02).

> Balance and coordination are also important. I think that by knowing how to stay balanced while moving demonstrates having much control over your body, your muscles, which means you can move well in a coordinated manner (FAM 10).

**Table 2. Descriptive data analysis.**

| Classification Tree | N Doc[a] | % Doc[b] | N Cod[c] | % Cod[d] | N Ext[e] | % Ext[f] |
|---|---|---|---|---|---|---|
| 1. Motor level | 42 | 100.00 | 196 | 15.62 | 588 | 15.35 |
| 1.1. Coordination and balance | 33 | 79.00 | 112 | 8.93 | 311 | 8.41 |
| 1.1.1. Motor-perceptual capacities | 33 | 79.00 | 112 | 8.93 | 311 | 8.41 |
| 1.2. Physical capacities | 19 | 45.00 | 52 | 4.14 | 190 | 5.14 |
| 1.2.1. Conditional capacities | 19 | 45.00 | 52 | 4.14 | 190 | 5.14 |
| 1.3. Basic and specific motor skills | 15 | 36.00 | 32 | 2.55 | 67 | 1.8 |
| 1.3.1. Basic motor skills | 9 | 21.00 | 18 | 1.43 | 35 | 0.94 |
| 1.3.2. Specific motor skills | 9 | 21.00 | 14 | 1.11 | 32 | 0.86 |
| 2. Cognitive level | 42 | 100.00 | 98 | 7.80 | 210 | 5.68 |
| 2.1. Perceived motor skill | 42 | 100.00 | 98 | 7.80 | 210 | 5.68 |
| 2.1.1. Perception of the fine motor skill | 29 | 69.00 | 48 | 3.82 | 100 | 2.70 |
| 2.1.2. Perception of the gross motor skill | 22 | 52.00 | 50 | 3.98 | 110 | 2.98 |
| 3. Affective-emotional level | 42 | 100.00 | 661 | 52.71 | 2132 | 57.71 |
| 3.1. Self-esteem | 42 | 100.00 | 271 | 21.61 | 976 | 26.42 |
| 3.1.1. Positively evaluating one's image and social relationship | 42 | 100.00 | 271 | 21.61 | 976 | 26.42 |
| 3.2. Anxiety | 42 | 42.00 | 173 | 13.79 | 529 | 14.32 |
| 3.2.1. Nervous attitudes and insecurities | 42 | 100.00 | 173 | 13.79 | 529 | 14.32 |
| 3.3. Commitment to learning | 42 | 100.00 | 217 | 17.3 | 497 | 13.45 |
| 3.3.1. Interest and positive implication | 42 | 100.00 | 181 | 14.43 | 446 | 12.07 |
| 3.3.2. Negative conducts and lack of interest | 18 | 43.00 | 26 | 2.07 | 51 | 1.38 |
| 4. Social level | 42 | 100.00 | 299 | 23.84 | 764 | 20.68 |
| 4.1. Promoting agents in the family and friendships | 42 | 100.00 | 163 | 12.99 | 390 | 10.55 |
| 4.1.1. Family members | 42 | 100.00 | 131 | 10.44 | 302 | 8.17 |
| 4.1.2. Friendships | 17 | 40.00 | 32 | 2.55 | 88 | 2.38 |
| 4.2. Promoting agents in the school environment | 34 | 81.00 | 97 | 7.73 | 250 | 6.76 |
| 4.2.1. PE Teachers | 17 | 40.00 | 30 | 2.39 | 60 | 1.62 |
| 4.2.2. Other teachers and management teams | 33 | 79.00 | 67 | 5.34 | 190 | 5.14 |
| 4.3. Promoting agents outside school | 11 | 26.00 | 14 | 1.11 | 30 | 0.81 |
| 4.3.1. Sport trainers/monitors | 11 | 26.00 | 14 | 1.11 | 30 | 0.81 |
| 4.4. Promoting agents in the institutional domain | 16 | 38.00 | 25 | 1.99 | 32 | 0.86 |
| 4.4.1. Institution and public administrations | 16 | 38.00 | 25 | 1.99 | 32 | 0.86 |

[a]Number of documents.

[b]Percentage of encoded documents.

[c]Number of times that the variable was encoded.

[d]Percentage of the total encoded references.

[e]Number of lines of the different coded information units.

[f]Percentage of the total encoded lines.

In category *1.2 Physical capacities*, family members believe that conditional capacities like speed or strength are related to good motor performance in the past: *I don't know how to explain it well, but it might be somewhat cultural because good movements have always been related to those who move quickly and strongly* (FAM 36).

Although teachers prioritize perceptive motor capacities over physical motor capacities, they also indicate the need to bear in mind certain capacities like strength, speed or resistance as determining factors of condition and physical activity:

These capacities help a child to more fluently perform individual motor actions by following a suitable technique. This is the first step or basis to move well in any physical activity that children wish to perform (MAE 15).

The results obtained in Category *1.3 Basic and specific motor skills* reveal that both teachers and family members understand skills as a very specific factor that is performed through learning and training. They point out that previous capacities are needed to develop them: *Because they allow children to apply these capacities that they master and to perform movements and skills well as a result* (FAM 32).

From my own training, to be able to handle objects, a general motor basis is first needed, which is provided by basic physical capacities or coordination, and these show good general movement. Then come more specific things like using objects, which do not necessarily involve making good movements (MAE 17).

Both social referents coincide in the fact that the specific motor skills required to use objects are more related to a competent child from the motor point of view: *Well, let's see. I think that the skills you need to control objects like balls need more performance*, or at least those more competent skills can be better handled (FAM 33).

## Cognitive level

Category *2.1 Perceived motor skill* refers to how good motor MC is perceived in the motor conducts that require using an object and those requiring only the body being employed by bearing in mind that better perceiving MC implies more motor performance possibilities. In line with this, teachers are the social referents who report a closer link between the motor and cognitive domains when demonstrating motor performance:

Actually objects are only a tool. What's really important is knowing your body and how to control it. Having a good self-concept is also very useful, but tends to develop later (MAE 16).

Family members and teachers also coincide in MC self-perception being more related to more complex skills requiring the use of objects. Both agents point out that the number of variables to take into account when using an object implies a higher degree of physical and cognitive complexity:

I think that using objects involves a higher degree of complexity, and needs the synergy between the mind and body to be greater with more awareness to coordinate more muscles and must, therefore, provide more responses (FAM 37).

Performing actions while using objects is more related to good movement because the involved cognitive capacity is greater as more variables must be taken into account. Good movement and also using an object involve a higher level, more difficulties and many other things (MAE 10).

Family members consider that using objects increases the degree of competence thanks to positive horizontal transfer, and in such a way that former learning about using objects facilitates learning to use other objects in similar contexts: *If you know how to control your body well, you can then play many sports where you have to use objects. Otherwise, it's very difficult*

(FAM 31). Teachers also refer to the importance of learning additional, superior or more complex contents to use the same object toward better motor performance:

> In my case, I think using objects helps to perform movement; I mean, the object helps movement to improve. So performing movements is beneficial. I also think that using an object shows that more training has been done. Therefore, movement is better and more complex (MAE 13).

## Affective-emotional level

Regarding this domain, teachers and family members also refer to the need to bear in mind the emotional-affective domain as part of developing MC: *I would go as far to say that the performance level improves, and very much so, in motivated students with high self-esteem and good self-concept* (MAE 06).

> I think that self-concept helps you to see yourself in a good light, but from a cognitive point of view, and self-esteem from your own emotional point of view, but also with others. This allows what is social and what is personal to be joined, which is important because both things are present when you practice physical activity (FAM 03).

In Category *3.1 Self-esteem*, perceiving the importance of this psychosocial variable is very marked, and impacts relevant development aspects like being predisposed to motor learning or social relationships with peers:

> I think it can have a positive effect. I mean, we always talk about it having a negative effect, but it can also be positive. I think that it can allow students to enjoy emotional well-being because they have their own people around them in a stable friendly environment (MAE 18).

Family members mainly attach more importance to characteristics in the context, and feel that those settings with a very low level of uncertainty, and knowing the people and the activity to be done, favor self-esteem: *It has a positive effect because children in a well-known setting can feel more accepted and more certainty when practising exercise, and they are set no limits* (FAM 05).

As for Category *3.2 Anxiety*, too much peer competition is a factor that is perceived more as possibly encouraging nervous conducts and high anxiety levels when performing motor conduct: *Yes, children sometimes feel pressure from their classmates, who expect or demand certain things being done. If children don't know how to do these things, they become nervous, or even paralyzed* (MAE 09). Here both teachers and family members point out that if these children feel ashamed or afraid is due to social pressure from peer groups, which results in worse motor performance: *If a relationship with classmates is bad, we think that they will laugh at us for failing. So performing activity with them will be like carrying a heavy burden, and it will require making more effort (FAM 08).*

> That, plus the emotional barrier of thinking about what classmates will think, thinking that you are going to fail, knowing that everyone will see you and might laugh at you, or being afraid of not rising to their expectations, or comparing yourself to them and then doing things worse than they do (MAE 02).

So like the self-esteem variable, family members are the main referents who understand the need for a known setting in which schoolchildren can positively develop from the emotional-motor point of view: *In relation to this, and as I previously mentioned, I think that self-esteem is very important in our relation to the activity and the setting* (FAM 08). Most family members show that anxiety and competition levels are on the increase and might trigger negative conducts:

> Having plenty of commonsense when they are very young is very important. As they grow older, there are more anxieties to outline and demonstrate. This leads to anger, bad conducts or more competitive conducts, which imply nothing positive (FAM 06).

Although teachers coincide with family members' perception, they also focus mainly on anxiety and motor performance because the motor level mostly triggers the nervous conducts that come into play while practising physical activity or not: *However, I think that a child who does not master these exercises will feel ashamed and afraid* (MAE 03).

Some agents also demand the need to jointly consider anxiety and self-esteem with cognitive variables to more globally understand students' MC by indicating causal relations among variables:

> I put myself in their place. Any child with poor motor performance can have low self-concept, and can also poorly perform in the motor domain, and his or her self-esteem can be low in these situations. So they feel more anxiety (FAM 19).

Another of the constructs to bear in mind in the affective-emotional domain is commitment to learning in both curricular and extracurricular terms (Category *3.3. Commitment to learning*). Schoolchildren of prepuberty age are generally well disposed to motor learning during school hours and while performing out-of-school activities.

The participants' perception is due mostly to the experience-type nature of the PE and practicing sports area in general. Learning by playful means and tools like games or sports allows schoolchildren to continue to practice them. Thus they entail physical, cognitive and favorable behavioral predisposition:

> Spending so much learning time in class, and these hours increasing as they move from one course to the next, performing any dynamic activity, especially during school hours, mean that motivation and fun will play an extremely important role in student learning (FAM 11).

They also point out the importance of group feeling when showing a positive attitude to learning because they outline a common project with their classmates and friends in which they feel responsible for and the shared members of: *I think that implication is greater in these cases because the pleasure of moving is joined to competition and being in a small peers group in which they share objectives and experiences* (MAE 08).

## Social level

About Category *4.1 Promoting agents in the family and friendships*, both family members and teachers coincide in mothers and fathers being the main agents who promote schoolchildren performing physical activity. This aspect has to do with them being mainly responsible for putting their child's name down to go to organizations or clubs where some kind of physical

activity is practiced, and encourage them to participate: *I think that family members shoulder much responsibility and must encourage young children to practice physical activity* (FAM 37).

Let's see. Parents must basically encourage this because, clearly without family support, it's not possible for a child to practice some type of physical activity, especially when the child is in a young development stage. All this involves parents making an effort and getting involved (MAE 14).

Family members also state that the family context must be the first setting in which models based on healthy habits have to appear. It must be a positive setting in which children are able to acquire a series of knowledge and attitudes from early ages:

If parents live a sedentary lifestyle and barely move, they will transmit this to their child. But if they are active and practice sport, from a very young age they will transmit the idea that sport is healthy and good for social relationships (FAM 08).

At school (*Category 4.2 Promoting agents in the school environment*), both teachers and management teams are responsible for promoting physical activity during school hours, improvement plans and the creation of spaces in education centers to supplement such promotion in the family context: *Working alongside teachers from schools is also essential because many family members do not know certain aspects of the evolutionary development or maturity related to physical activity* (FAM 09).

PE teachers also play a key role given the nature of the knowledge area that they teach, where social, cognitive and motor contents merge: *Having teachers who get involved, are aware and ready to work these PE contents by taking into account age, the context and suitable materials is very important* (FAM 07).

Bearing in mind other social contexts, like sport clubs or physical activities not practiced at school (*Category 4.3 Promoting agents outside school*), the participants point out that both trainers and monitors are the agents who least influence the promotion of physical activity because of the competitive nature with which they face out-of-school activities: *In contrast, clubs and sports organizations do not see this in the same way; they focus on the obtained result* (MAE 08).

But monitors and trainers sometimes also take out-of-school activities as competitive. It's fine to compete at times and, obviously, all sports have their competitive side, but I think that they should steer sport more toward the social benefits you mention (FAM 15).

Finally, the results obtained in Category *4.4 Promoting agents in the institutional domain* evidence the need to increase the number of resources and economic investment to create opportunities in the public area so that schoolchildren can practice physical activity: *Institutions must facilitate opportunities and more public spaces so everyone feels they participate* (MAE 15).

Family members also identify the need to resort to a private area to the detriment of the public area so that schoolchildren can perform physical activity: *Although public institutions also back the benefits of sport and food, they don't have many offers. You must almost always go to a private center if you're looking for a quality sport service* (FAM 13). Family members also stress the need to update and revise education laws in school hours terms: *Starting with education lawmakers, bearing in mind that they need to be included in the curriculum, as well as a suitable number of hours to carry them out* (MAE 07).

Table 3 shows all the quotations used in the interpretative analysis.

**Table 3. Quotations used in the interpretative analysis.**

| Subtheme | Reference | Participant |
|---|---|---|
| 1.1.1. Motor-perceptual capacities | I think that balance is basic to move well both statically and while moving because balance allows the body to be erected and to adopt an optimum position to perform movement | MAE02 |
| | That's not all because, on the other hand, to well coordinate movements, or to combine movements with others with plasticity, it is necessary to consider the exterior stimuli that do not form part of our body so our body can do what we really want it to do | MAE 04 |
| | Balance and coordination are also important. I think that by knowing how to stay balanced while moving demonstrates having much control over your body, your muscles, which means you can move well in a coordinated manner | FAM 10 |
| | I think that good motor performance is mainly related to coordination because I consider coordination to be the cogs that enable other capacities to work properly | FAM 15 |
| 1.2.1. Conditional capacities | These capacities help a child to more fluently perform individual motor actions by following a suitable technique. This is the first step or basis to move well in any physical activity that children wish to perform | MAE 15 |
| | I don't know how to explain it well, but it might be somewhat cultural because good movements have always been related to those who move quickly and strongly | FAM 36 |
| 1.3.1. Basic motor skills | From my own training, to be able to handle objects, a general motor basis is first needed, which is provided by basic physical capacities or coordination, and these show good general movement. Then come more specific things like using objects, which do not necessarily involve making good movements | MAE 17 |
| | Because they allow children to apply these capacities that they master and to perform movements and skills well as a result | FAM 32 |
| 1.3.2. Specific motor skills | Well, let's see. I think that the skills you need to control objects like balls need more performance, or at least those more competent skills can be better handled | FAM 33 |
| 2.1.1. Perception of the fine motor skill | Performing actions while using objects is more related to good movement because the involved cognitive capacity is greater as more variables must be taken into account. Good movement and also using an object involve a higher level, more difficulties and many other things | MAE 10 |
| | In my case, I think using objects helps to perform movement; I mean, the object helps movement to improve. So performing movements is beneficial. I also think that using an object shows that more training has been done. Therefore, movement is better and more complex | MAE 13 |
| | Actually objects are only a tool. What's really important is knowing your body and how to control it. Having a good self-concept is also very useful, but tends to develop later | MAE 16 |
| | I think that using objects involves a higher degree of complexity, and needs the synergy between the mind and body to be greater with more awareness to coordinate more muscles and must, therefore, provide more responses | FAM 37 |
| 2.1.2. Perception of the gross motor skill | If you know how to control your body well, you can then play many sports where you have to use objects. Otherwise, it's very difficult | FAM 31 |

(*Continued*)

**Table 3.** (Continued)

| Subtheme | Reference | Participant |
|---|---|---|
| 3.1.1. Positively evaluating one's image and social relationship | I would go as far to say that the performance level improves, and very much so, in motivated students with high self-esteem and good self-concept | MAE 06 |
| | I think it can have a positive effect. I mean, we always talk about it having a negative effect, but it can also be positive. I think that it can allow students to enjoy emotional well-being because they have their own people around them in a stable friendly environment | MAE 18 |
| | I think that self-concept helps you to see yourself in a good light, but from a cognitive point of view, and self-esteem from your own emotional point of view, but also with others. This allows what is social and what is personal to be joined, which is important because both things are present when you practice physical activity | FAM 03 |
| | It has a positive effect because children in a well-known setting can feel more accepted and more certainty when practicing exercise, and they are set no limits | FAM 05 |
| 3.2.1. Nervous attitudes and insecurities | That, plus the emotional barrier of thinking about what classmates will think, thinking that you are going to fail, knowing that everyone will see you and might laugh at you, or being afraid of not rising to their expectations, or comparing yourself to them and then doing things worse than they do | MAE 02 |
| | However, I think that a child who does not master these exercises will feel ashamed and afraid | MAE 03 |
| | Yes, children sometimes feel pressure from their classmates, who expect or demand certain things being done. If children don't know how to do these things, they become nervous, or even paralysed | MAE 09 |
| | Having plenty of common-sense when they are very young is very important. As they grow older, there are more anxieties to outline and demonstrate. This leads to anger, bad conducts or more competitive conducts, which imply nothing positive | FAM 06 |
| | In relation to this, and as I previously mentioned, I think that self-esteem is very important in our relation to the activity and the setting | FAM 08 |
| | If a relationship with classmates is bad, we think that they will laugh at us for failing. So performing activity with them will be like carrying a heavy burden, and it will require making more effort | FAM 08 |
| | I put myself in their place. Any child with poor motor performance can have low self-concept, and can also poorly perform in the motor domain, and his or her self-esteem can be low in these situations. So they feel more anxiety | FAM 19 |
| 3.3.1. Interest and positive implication | I think that implication is greater in these cases because the pleasure of moving is joined to competition and being in a small peers group in which they share objectives and experiences | MAE 08 |
| | Spending so much learning time in class, and these hours increasing as they move from one course to the next, performing any dynamic activity, especially during school hours, mean that motivation and fun will play an extremely important role in student learning | FAM 11 |

(*Continued*)

**Table 3.** (Continued)

| Subtheme | Reference | Participant |
|---|---|---|
| 4.1.1. Relatives | Let's see. Parents must basically encourage this because, clearly without family support, it's not possible for a child to practice some type of physical activity, especially when the child is in a young development stage. All this involves parents making an effort and getting involved | MAE 14 |
| | If parents live a sedentary lifestyle and barely move, they will transmit this to their child. But if they are active and practice sport, from a very young age they will transmit the idea that sport is healthy and good for social relations | FAM 08 |
| | I think that families shoulder much responsibility and must encourage young children to practice physical activity | FAM 37 |
| 4.2.1. PE Teachers | Having teachers who get involved, are aware and ready to work these PE contents by taking into account age, the context and suitable materials is very important | FAM 07 |
| | Working alongside teachers from schools is also essential because many families do not know certain aspects of the evolutionary development or maturity related to physical activity | FAM 09 |
| 4.3.1. Sport trainers/monitors | In contrast, clubs and sports organisations do not see this in the same way, they focus on the result obtained | MAE 08 |
| | But also with monitors and trainers sometimes take out-of-school activities as competitive. It's fine to compete at times and, obviously, all sports have their competitive side, but I think that they should steer sport more towards the social benefits you mention | FAM 15 |
| 4.4.1. Institution and public administrations | Starting with education lawmakers, bearing in mind that they need to be included in the curriculum, as well as a suitable number of hours to carry them out | MAE 07 |
| | Institutions must facilitate opportunities and more public spaces so everyone feels they participate | MAE 15 |
| | Although public institutions also back the benefits of sport and food, they don't have many offers. You must almost always go to a private centre if you're looking for a quality sport service | FAM 13 |

## Discussion

The main objective of this study is to analyze the perception of social referents like family members and teachers of the implication of motor, social, emotional and cognitive variables in the multidimensional MC construct in the pre-adolescent stage. This includes an evolutionary component that contributes to schoolchildren's overall development. Hence these social referents' perceptions contribute to better understand which types of variables are present and how they come over [39]. Therefore, the main objective of the present study is to know social referents' perception of MC from a qualitative domain by proving that our results fall in line with other quantitative-type studies. In this way, it is shown that family members and teachers believe that the motor domain is important, but also state that the affective-emotional domain is the determining factor to understand the MC concept and to promote its development in the pre-adolescent stage. They also think that family, members are the main promoting agent in situations in which MC can develop.

In the motor domain, both family members and teachers believe that coordination is the main capacity related to schoolchildren's good MC. This aspect agrees what Ruiz-Pérez et al. [40] set out, who established the need to make coordination-based resources available to perform motor tasks and to meet the motor objective. Drenowatz and Dreier [41] also state that

to be motor-competent, schoolchildren have to develop a series of coordination synergies or structures that enable them to have a controllable motor system and to more efficiently perform actions. Cañizares and Carbonero [42] also state the importance of such learning for children and pre-adolescents in their first schooling stage to allow them to create a motor structure based on coordinator motor patterns, and on advancing toward the learning and development of specific skills that imply employing objects.

Apart from coordination capacity, other cognitive variables must be considered, such as self-perceiving MC [43]. Both family members and teachers state that schoolchildren's self-perception of their own performance is very important when performing motor tasks. They point out that specific motor skills appear to be those that are more related to good MC because they involve a higher level of difficulty when being performed and involve more elaborate learning [44].

If schoolchildren acquire a positive self-concept, it allows them to feel sure in confidence terms about safely carrying out motor tasks, which positively influences their own performance while doing them [45]. This aspect has also been evidenced by authors like Estevan and Barnett [10], who understand that if schoolchildren acquire lived positive motor experiences, it will allow a better image of themselves to be created in relation to the efficiency with which they perform motor gestures. So it is necessary for schoolchildren to engage in many positive motor situations inside and outside school to increase the number of motor experiences.

Likewise, many of the physical activities practiced both inside and outside school involve handling mobile elements and objects [46]. This implies engaging schoolchildren in performing a task that centers on not only motor action from the analytical viewpoint, but also the performed motor conduct implies more cognitive implication because more variables must be considered (object, adversaries, partners, space, etc.). Barnett et al. [47] corroborate that those skills involving the control of objects are related to better motor performance, and also predict schoolchildren's perceived MC. Thus developing specific motor skills using objects is related to more self-perceived MC.

On self-esteem, both social referents consider that it is one of the most relevant variables to suitably develop MC. One of the main tasks lies in working on self-esteem from all areas by means of knowledge and accepting our own body, ourselves, and our motor capacities, possibilities and limitations [48]. Family members attach much importance to the need for a stable environment to be known by students to develop their MC in, with no uncertainty from the social point of view, and knowing the classmates with whom they are to relate. This allows students to be more willing to perform physical activity and their emotional stability will grow. Cañizares and Carbonero [49] state that performing physical activity in a known setting, with little uncertainty and surrounded by people who are close to them from their context, helps students to feel more self-esteem when performing motor conducts because they perceive a more comforting environment. So having a certain degree of trust in performing a task, and knowing the environment and the conditions where the task is to be done, allow students to feel more willing to perform that task [50].

On anxiety, teachers perceive the notion that the degree of competition rises with increasing age, and the number of anxiety-related and other negative conducts grows with which people intend to reach a certain social status or hierarchy in the group-class. Conversely, family members mention that anxiety strongly impacts MC development because those students who carry out motor performance feel individual failure or nervousness less when interacting with classmates. We also find that manifesting anxiety can involve paying attention to oneself and other classmates. In line with this, Cox [51] points out that the joint existence of cognitive anxiety and somatic anxiety in pre-adolescent pupils is possible in the school context. So feeling cognitive anxiety or concerns about failure and the fear of social negative evaluations implies

subjects having more somatic responses [52]. Indeed excess peer competition and the social pressure of performing a series of motor actions in a social group also imply certain nervous and insecure conducts coming into play, which encourage lack of interest in practicing physical activity [53].

The men and women students with a MC that is negatively valued by either themselves or people who are meaningful to them also make an inevitable comparison to, and feel social pressure from, classmates. Hence the need to create suitable, relevant and feasible contexts in which schoolchildren can develop by creating conditions that ensure them successfully performing tasks [54]. These conditions are shaped by methodological principles, such as co-education and paying attention to diversity, by bearing in mind particular characteristics and avoiding isolation and social rejection [55]. So these learning environments must be focused on from an education perspective and to prolong the education action carried out in education centers [17].

Regarding commitment, both family members and teachers coincide that the willingness to perform activities or studying subjects in which MC is developed is generally positive. Nevertheless, they also indicate that this commitment varies according to the type of methodology or the related characteristics of the activities to be performed. Indeed commitment and interest lower as students come closer to adolescence because of changes in evolutionary maturity rhythms, being exposed to motor tasks and the social environment's influence [56, 57]. This means that the way work is done during both PE sessions and out-of-school activities must be dealt with from a teaching process based on innovative and active methodologies [58]. Hence the need to shape learning situations that stimulate the group's motor practice in general and include the different promoted learning rhythms. This approach is based on successful pedagogy, which allows students' intrinsic motivation to increase and, hence, their intention to be more physically involved [59].

In the social domain, close referents are responsible mostly for encouraging and promoting schoolchildren's motor practice during school hours and outside school [60]. Both family members and teachers refer to the family circle being the agent that promotes contexts in which to develop MC to a greater extent. Here the main justification lies in family members being responsible for promoting and transmitting models of healthy habits, and participating in out-of-school activities [61]. Accordingly, Márquez-Cervantes and Gaeta-González [62] establish the stronger influence of the closest social setting and that with the most contact in which schoolchildren live because of it being a context in which the typical attitudes, knowledge and behaviors of the culture surrounding schoolchildren are learned. This fact is due to pre-adolescence being a period during which schoolchildren carry out learning based on observing models, and in such a way that if schoolchildren perceive the consequences of a given behavior in other close referents, they amend their own [63].

Albeit to a lesser extent, both teachers and family members also point out that the education center as an organizational institution is responsible for promoting contexts in which students develop their MC. As such, it is important that teachers also encourage and promote usual physical activity by facilitating schoolchildren acquiring healthy and active habits [27, 64, 65]. Ruiz-Pérez [66] states that the set of education institution members is responsible for evaluating schoolchildren's MC by covering any possible education needs or difficulties that may arise. Nonetheless for teachers on the whole, family members refer particularly to the role played by PE teachers owing to the motor and experience-based nature of their knowledge area.

In light of all this, authors like Benedito and Parcerisa [67] contemplate that the MC character falls in line with a global consideration, but differs from those positivists who refer only to the concept from a motor approach. To do so, an MC concept based on a socio-constructivist

or postcognitive approach is considered that attaches more relevance to social, cognitive and emotional elements, and also to the relevance of the context, the setting and social interaction to develop this competence [2]. Along the same lines, Águila-Soto and López-Vargas [68] state that PE must move away from a mechanistic and technical vision of motor development and be backed by an approach that stimulates certain aspects like affective balance or sociability.

This study has certain limitations. The number of participants in both groups (teachers and family members) may condition the obtained results. However, and taking into account the saturation of the sample in terms of its capacity to provide information to understand the studied phenomenon and to respond to the study objectives [69], we could consider that the employed sample could respond to the methodological requirements related to rigor. In this sense, and in accordance with previous studies [70, 71], using a form of code frequency counts to assess saturation, which involved counting codes in a set of transcripts until no more codes were identified and randomizing the order of data to assess the influence of sequential bias on saturation, it was determined that we needed a minimum of 20 individualized interviews with family members and three focus groups composed of teachers. Moreover, another limitation was to configure our study sample which centred on teachers and family members. As possible future research lines, it would be worthwhile including and considering other schoolchildren's close social referents, such as their classmates, sport groups and friendships. A study that brings together objects from a multimethod perspective could also be contemplated. Hence the importance of profoundly examining social referents' perception of developing schoolchildren's MC because it allows us to know how the different involved variables emerge and are stressed to a greater extent, and how they are transformed to, therefore, set the bases to better know how to cultivate them. All this must be done by taking into account that MC undergoes lifelong development and develops integrally according to conditioning environmental factors.

## Conclusions

The perception of educational social referents like family members and teachers is most important when attempting to deal with and look closely at those variables involved in the MC concept to a greater extent given the influence on children. The family transmits a critical impact on their children's motor and social development by helping them to create a series of positive attitudes toward physical activity and a personal vision based on outlines shaped by parents. Teachers also play an important psychosocial role for physical activity by stimulating different physico-sport activities being practiced.

Therefore, joint collaboration between family members and teachers as education community members seems necessary to promote schoolchildren to develop their MC both inside and outside schools. As for possible ways of collaboration, on the one hand, there is the possibility of applying a significant transfer of the contents related to physical activity worked on in the school environment with sports institutions or organizations that are close to the schools to, thus, allow students to continue to promote adherence to physical practice with a low uncertainty level; on the other hand, establishing alliances and collaborations between the school center and organizations, institutions or sports clubs in the nearby environment allows family members to have a more favorable and facilitating context to find opportunities in which schoolchildren can continue their motor development.

## Supporting information

**S1 File. Classification tree.**
(DOC)

**S2 File. Semistructured interview script and discussion group.**
(DOC)

**S3 File. Consolidated criteria for reporting qualitative studies (COREQ): 32-item checklist.**
(DOC)

## Acknowledgments

The authors wish to thank all the teachers and family members for their involvement in this study.

Manuel Lizalde-Gil passed away before the submission of the final version of this manuscript. Juan Carlos Bustamante accepts responsibility for the integrity and validity of the data collected and analyzed.

## Author Contributions

**Conceptualization:** Manuel Segura-Berges, Carlos Peñarrubia-Lozano, Juan Carlos Bustamante.

**Formal analysis:** Manuel Segura-Berges.

**Funding acquisition:** Carlos Peñarrubia-Lozano.

**Investigation:** Manuel Segura-Berges, Manuel Lizalde-Gil.

**Methodology:** Manuel Segura-Berges, Carlos Peñarrubia-Lozano, Juan Carlos Bustamante.

**Project administration:** Juan Carlos Bustamante.

**Supervision:** Carlos Peñarrubia-Lozano, Juan Carlos Bustamante.

**Validation:** Manuel Segura-Berges, Manuel Lizalde-Gil.

**Writing – review & editing:** Carlos Peñarrubia-Lozano, Manuel Lizalde-Gil, Juan Carlos Bustamante.

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
