## [Decision Letter · Decision Letter 0]

19 Jul 2022

PONE-D-22-00018

Qualitative analysis from the social referents perspective of the multidimensional construct of schoolchildren’s motor competence

PLOS ONE

Dear Dr. Bustamante,

Thank you for submitting your manuscript to PLOS ONE. After careful consideration, we feel that it has merit but does not fully meet PLOS ONE’s publication criteria as it currently stands. Therefore, we invite you to submit a revised version of the manuscript that addresses the points raised during the review process.

Please note that we have only been able to secure a single reviewer to assess your manuscript. We are issuing a decision on your manuscript at this point to prevent further delays in the evaluation of your manuscript. Please be aware that the editor who handles your revised manuscript might find it necessary to invite additional reviewers to assess this work once the revised manuscript is submitted. However, we will aim to proceed on the basis of this single review if possible. 

The reviewer raises several concerns about the study design and request clarifications. Their comments are attached below. Would you please revise the manuscript to address their questions?

We look forward to receiving your revised manuscript.

Kind regards,

Thomas Tischer

Staff Editor

PLOS ONE

Journal Requirements:

3. Please change "female” or "male" to "woman” or "man" as appropriate, when used as a noun (see for instance https://apastyle.apa.org/style-grammar-guidelines/bias-free-language/gender).

5. Please amend your manuscript to include your abstract after the title page.

Additional Editor Comments (if provided):

We noticed that limitations of the study, like the low number of participants or potential sources of bias, have not been discussed. Please include a paragraph addressing these concerns.

Reviewers' comments:

Reviewer's Responses to Questions

**Comments to the Author**

1. Is the manuscript technically sound, and do the data support the conclusions?

Reviewer #1: Yes

2. Has the statistical analysis been performed appropriately and rigorously? 

Reviewer #1: Yes

3. Have the authors made all data underlying the findings in their manuscript fully available?

Reviewer #1: Yes

4. Is the manuscript presented in an intelligible fashion and written in standard English?

Reviewer #1: Yes

5. Review Comments to the Author

Reviewer #1: First, I would like to congratulate the authors for the efforts done in the manuscript. I think that it is very interesting and It will be a good starting point for other researchers that focus on this field of motor competence. The paper is well structured although the English grammar must be reviewed. For example, “conducted” instead of “carry out”. The introduction is well written but a little short. I miss more information related to previous research motor competence related. Which evidence exist? It would be important to highlight why this study is new and why it is important that is published. What does this study add to the already published before?

About the methodology, I have some concerns that I would like to ask to the authors:

• Why did you choose 4, 5 and 6 grade?

• Why did you choose an ethnographic approach? Justify.

• How did you check the grade in which the teachers work?

• How did you create the interview? Was it validated? Was it reviewed by experts?

About the results, it would be interesting to show a table with examples of quotations of each subtheme. In the discussion section, you mention one quotation of each theme and subtheme. I would recommend using more quotations. Finally, the conclusions must be written without references, please remove. Moreover, the strengths, limitations and prospective must be changed to the end of discussion. And last, when you say “joint collaboration between families and teachers as education community members seems necessary to promote schoolchildren to develop their motor competence both inside and outside schools”, please could you make some examples in that sense?

6. PLOS authors have the option to publish the peer review history of their article (what does this mean?). If published, this will include your full peer review and any attached files.

Reviewer #1: No

---

## [Author Response · Author response to Decision Letter 0]

2 Sep 2022

Dear Editor.

It is a pleasure for us to receive the opportunity to improve our manuscript by taking into account the Editor’s useful comments and the Reviewer’s interesting comments. 

We would like to thank both the Editor for making the revision of our manuscript possible and the Reviewer for all his/her comments and suggestions, which have undoubtedly improved the writing and quality of the study. Below we list our replies to each comment and suggestion made. Changes are highlighted and included in the manuscript.

EDITOR´S COMMENTS

Response: We have ensured that the revised version meets PLOS ONE’s style requirements.

Response: In the ethics statement in the Methods and the online submission information we have specified what type of consent form we obtained. It was obtained in writing. Likewise in the Data collection procedure section, we have indicated that this informed consent was again recalled verbally at the beginning of both the interviews (family members) and focus groups (teachers). Our study did not include minors.

3. Please change "female" or "male" to "woman" or "man" as appropriate, when used as a noun (see for instance https://apastyle.apa.org/style-grammar-guidelines/bias-free-language/gender).

Response: We have modified “female” or “male” to “woman” or “man”.

Response: We have specified where the dataset and transcripts that underlie the results described in our manuscript can be found. The dataset has been deposited in the Figshare data repository.

5. Please amend your manuscript to include your abstract after the title page.

Response: We have included the Abstract after the title page.

Additional Editor Comments:

We noticed that limitations of the study, like the low number of participants or potential sources of bias, have not been discussed. Please include a paragraph addressing these concerns.

Response: These concerns have been discussed, and we have also added some methodological aspects regarding saturation.

REVIEWER’S COMMENTS

Reviewer #1: First, I would like to congratulate the authors for the efforts done in the manuscript. I think that it is very interesting and It will be a good starting point for other researchers that focus on this field of motor competence. The paper is well structured although the English grammar must be reviewed. For example, "conducted" instead of "carry out".

The introduction is well written but a little short. I miss more information related to previous research motor competence related. Which evidence exist? It would be important to highlight why this study is new and why it is important that is published. What does this study add to the already published before?

Response: We would like to thank the Reviewer for his/her comments, which improved the manuscript. The manuscript has been revised by a native English proofreader before submission. However, a second revision has been conducted (see the certificate below). Likewise, the proposed change has been made. 

Regarding the Introduction, information related to motor competence research has been added. In this way, the different variables that define the new model of motor competence and the relations found between them have been shown. With all this, the novelty of this work lies in not only the relations between the variables, but also in the incorporation of social agents into the study because they are considered a key element in the development of motor competence in the Primary Education stage on which this study focuses. 

About the methodology, I have some concerns that I would like to ask to the authors:

• Why did you choose 4, 5 and 6 grade?

Response: In the Spanish education system, grades 4, 5 and 6 correspond to 9- to 12-year olds. As stated in the Introduction, previous studies refer to the importance of the perception of schoolchildren’s significant and close people to deal with these children’s motor competence concept at pre-adolescent ages between 9 and 12 years old. Moreover according to previous studies, the stage between 9e and 12 years old is when the greatest motor development occurs from the evolutionary and structural points of view. It is also a time when cognitive and emotional aspects are especially important in personality development, when physical and psychological changes take place which, in turn, transform their biology, behavior and social relationships.

• Why did you choose an ethnographic approach? Justify.

Response: We agree with the Reviewer that this issue needs clarifying. We have adjusted this concern in the manuscript.

• How did you check the grade in which the teachers work?

Response: A convenience sampling process was used to set up the sample and age was one of the variables included in the inclusion criteria. Thus when presenting the informed consent form, the participants were asked to indicate the course in which they were taught.

 • How did you create the interview? Was it validated? Was it reviewed by experts?

Response: The entire categorization process (generation of the system of categories at the three indicated levels, together with their definitions; S1 File) was reviewed by an external person, an expert in qualitative methodology with training and experience in the Physical Education area. Once the system of categories was revised, the semistructured interview and discussion group scripts were prepared in an attempt to generate questions that would allow all the indicators (the system’s third level of concreteness) to be agglutinated in the answers provided by the different groups of participants. These questions were also reviewed by the expert in qualitative methodology, whose contributions (change in the order and wording of some questions) were considered. In the S2 File, we present the final version of the script used in the study. All this information has been included in the Data collection procedure section.

About the results, it would be interesting to show a table with examples of quotations of each subtheme.

Response: Table 3 has been added, which includes the different employed quotations.

In the discussion section, you mention one quotation of each theme and subtheme. I would recommend using more quotations.

Response: We did not include quotations in the Discussion section. Quotations were included in the Results section, but more references have been included.

Finally, the conclusions must be written without references, please remove.

Response: The requested change has been made. Thank you very much. Likewise, the references have been included in the Discussion section.

Moreover, the strengths, limitations and prospective must be changed to the end of discussion.

Response: The requested change has been made by adding the specific comment on the limitations suggested by the journal's Editors.

And last, when you say "joint collaboration between families and teachers as education community members seems necessary to promote schoolchildren to develop their motor competence both inside and outside schools", please could you make some examples in that sense?

Response: Some examples have been included in the Conclusions section. 

As you can see, we have considered all the advice offered and concerns raised by the Editor and Reviewer. 

We hope that you receive our change proposals optimistically with a view to publish this manuscript.

Yours faithfully,

The authors.

---

## [Editor Report · Decision Letter 1]

12 Sep 2022

Qualitative analysis from the social referents perspective of the multidimensional construct of schoolchildren’s motor competence

PONE-D-22-00018R1

Dear Dr. Bustamante,

We’re pleased to inform you that your manuscript has been judged scientifically suitable for publication and will be formally accepted for publication once it meets all outstanding technical requirements.

Kind regards,

Ender Senel, PhD

Academic Editor

PLOS ONE
---

## [Editor Report · Acceptance letter]

8 Dec 2022

PONE-D-22-00018R1 

Qualitative analysis from the social referents perspective of the multidimensional construct of schoolchildren’s motor competence 

Dear Dr. Bustamante:

I'm pleased to inform you that your manuscript has been deemed suitable for publication in PLOS ONE. Congratulations! Your manuscript is now with our production department. 

Kind regards, 

on behalf of

Dr. Ender Senel 

Academic Editor

PLOS ONE